# Key performance indicators of COVID-19 contact tracing in Belgium from September 2020 to December 2021

**Cécile Kremer**[1☯]*, **Lander Willem**[2,3☯], **Jorden Boone**[4], **Wouter Arrazola de Oñate**[5,6], **Naïma Hammami**[7], **Christel Faes**[1], **Niel Hens**[1,2]

**1** Interuniversity Institute for Biostatistics and statistical Bioinformatics, Data Science Institute, Hasselt University, Hasselt, Belgium, **2** Centre for Health Economics Research and Modelling Infectious Diseases, Vaccine and Infectious Disease Institute, University of Antwerp, Antwerp, Belgium, **3** Family Medicine and Population Health, University of Antwerp, Antwerp, Belgium, **4** KPMG Advisory, Public Sector Practice, Zaventem, Belgium, **5** Belgian Lung and Tuberculosis Association, Brussels, Belgium, **6** Flemish Association for Respiratory Health and Tuberculosis, Leuven, Belgium, **7** Department of Infectious Disease Prevention and Control, Department of Care, Flemish Region, Brussels, Belgium

☯ These authors contributed equally to this work.
* cecile.kremer@uhasselt.be

**Data Availability Statement:** The R code used for data manipulation and analysis is available on Github (https://github.com/cecilekremer/tracingKPI).

## Abstract

The goal of tracing, testing, and quarantining contacts of infected individuals is to contain the spread of infectious diseases, a strategy widely used during the COVID-19 pandemic. However, limited research exists on the effectiveness of contact tracing, especially with regard to key performance indicators (KPIs), such as the proportion of cases arising from previously identified contacts. In our study, we analyzed contact tracing data from Belgium collected between September 2020 and December 2021 to assess the impact of contact tracing on SARS-CoV-2 transmission and understand its characteristics. Among confirmed cases involved in contact tracing in the Flemish and Brussels-Capital regions, 19.1% were previously identified as close contacts and were aware of prior exposure. These cases, referred to as 'known' to contact tracing operators, reported on average fewer close contacts compared to newly identified individuals (0.80 versus 1.05), resulting in fewer secondary cases (0.23 versus 0.28). Additionally, we calculated the secondary attack rate, representing infections per contact, which was on average lower for the 'known' cases (0.22 versus 0.25) between December 2020 and August 2021. These findings indicate the effectiveness of contact tracing in Belgium in reducing SARS-CoV-2 transmission. Although we were unable to quantify the exact number of prevented cases, our findings emphasize the importance of contact tracing as a public health measure. In addition, contact tracing data provide indications of potential shifts in transmission patterns among different age groups associated with emerging variants of concern and increasing vaccination rates.

## Introduction

Coronavirus disease 2019 (COVID-19), caused by the severe acute respiratory syndrome coronavirus 2 (SARS-CoV-2), emerged in December 2019 and led to a global cumulative number

**Funding:** Part of this project was funded by the Flemish Government's Department of Care, which also provided the data. The funding source had no involvement in study design, analysis, nor in the decision to submit the paper for publication.

**Competing interests:** Naima Hammami is an employee of the the Flemish Government's Department of Care and Jorden Boone has been working as a consultant for this agency during the COVID-19 pandemic. Niel Hens declares that the Universities of Antwerp and Hasselt have received funding for advisory boards and research projects of MSD, GSK, JnJ, Pfizer outside the proposed work. Niel Hens has not received any personal remuneration related to this work. The other authors declare that they have no competing interests. This does not alter our adherence to PLOS ONE policies on sharing data and materials.

of confirmed cases of more than 598 million and a cumulative number of reported deaths over 6.4 million at the end of August 2022 [1]. To maintain the healthcare system, non-pharmaceutical interventions (NPIs) have been adopted during the pandemic in many countries. In the philosophy of having person-tailored quarantine measures, COVID-19 contact tracing was implemented in several countries. Active case finding and swift contact tracing have been effective for tuberculosis prevention and early treatment for many years [2, 3]. However, the scale required for COVID-19 was unprecedented in many European countries. Effective contact tracing can, when carried out quickly, prevent onward transmission from newly infected individuals and their possibly infected contacts [4]. Hence timely notification of possible exposure to SARS-CoV-2 is essential to break transmission chains, as indicated by several mathematical modeling studies investigating the impact of contact tracing [5–9]. However, there is a need for empirical studies on the effectiveness of contact tracing [10, 11].

In the absence of a reference period or region, it is not trivial to quantify the effectiveness of contact tracing in terms of the number of cases and deaths averted [12]. However, the potential impact of contact tracing on onward transmission can be evaluated using several key performance indicators (KPIs). The European Center for Disease Control (ECDC) and the World Health Organization (WHO) have proposed the proportion of new cases arising from known contacts as an indication of the quality and completeness of the contact tracing system, by pointing out the capacity of the system to identify all potential cases [13, 14]. As such, a higher proportion of cases that were previously identified as contact indicates that more individuals at risk of infection are reached by contact tracing. Other KPIs include delay distributions, such as the time between the onset of symptoms, clinical tests, and quarantining exposed contacts of an index case. Contact tracing also enhances active surveillance and provides unique data to estimate transmission characteristics, such as secondary attack rates, risk factors associated with infection, and important parameters that determine the speed at which a virus is spreading.

In this study, we used contact tracing data from Belgium (Flemish and Brussels-Capital region) collected between September 2020 and December 2021 to investigate (1) the proportion of all confirmed SARS-CoV-2 cases that were previously traced as a high- or low-risk contact, i.e. 'known' cases, (2) the number of traced contacts for 'known' and 'new' cases, (3) the number of potential secondary cases for 'known' and 'new' cases, (4) the ratio of potential infections over contacts, i.e. secondary attack rate, and (5) delay distributions. In addition, we investigated social contact and transmission patterns based on these data.

## Materials and methods

### Description of the contact tracing system

On 7 May 2020, the Flemish Government's Department of Care started with large-scale contact tracing in order to identify risk contacts of confirmed SARS-CoV-2 cases and alert them of possible exposure to the virus. This coincided with the deconfinement of several restrictive measures from the first lockdown in Belgium. An index case was defined as a confirmed case whose detection initiated a contact tracing event. Index cases were queried about any contacts that occurred from within the two days before symptom onset, or within the two days before their positive SARS-CoV-2 test result in case of no symptoms. Inquiries were made via telephone by call agents at centralized locations. Contacts reported by index cases were classified as high- or low-risk. High-risk contacts (HRC) were defined as physical contacts, or non-physical contacts with a duration of more than 15 minutes within a 1.5 m distance without correct use of face masks. Other contacts not meeting these conditions were deemed low-risk contacts (LRC). Testing and quarantine measures for HRC varied throughout the study period, and

LRC were instructed to heighten their vigilance and undergo testing if they exhibited any symptoms [15].

Interview data from the confirmed COVID-19 cases contained demographic information, an encrypted national registry number (NRN), the number of unique persons that an index case had met in the last 48 hours before their positive test or symptom onset (i.e. reported contacts), the number of reported contacts that were registered in the contact tracing database (i.e. registered contacts), and information on the registered contacts that were effectively reached by the contact tracing (i.e. traced contacts). Registered contacts that could not be reached and incomplete contact information hampered the number of contacts traced. Additionally, contacts that were reported by more than one index case were only registered once. Household members who were already in quarantine or isolation were usually not registered as contacts to trace. Note that these two exclusions were adopted during an update of the contact tracing protocol early in the study period, resulting in not all reported contacts being registered. Reported contacts for which no name could be provided were also not registered. Interview data for LRC and HRC included demographic data and an encrypted NRN. The latter allowed us to identify 'known' index cases after testing positive. If the NRN was missing, contacts were excluded from the analyses. Contacts that could not be linked to an index case due to missing information (e.g. when the reported index case is not a resident of the included regions) were excluded from the analyses. If contacts were linked to the same index case more than once, the duplicated information was removed. In collectivities such as hospitals, schools, and nursing homes, only individual contacts occurring outside these settings were available. Contacts occurring within a collectivity were followed up by the responsible medical services, these data were not available to us and hence not included in our study.

In the fall of 2020, Belgium experienced a severe second wave of COVID-19 during which the test capacity was under great pressure, resulting in a temporary change in the test strategy. The workload for the contact tracing also increased, and as a result only index cases were called during that time, whereas HRC were informed via text message. More specifically, between 16 October and 2 November 2020, text messages were sent to most HRC, while LRC were not traced at all. As a consequence, the NRN was not recorded for most HRC and LRC and it was not possible to identify these contacts as 'known' index cases during September–November 2020. After the summer of 2021, the contact tracing strategy changed to an approach where index cases who had attended some sort of gathering were queried in more detail when there was at least one other index linked to the same gathering or when the index case mentioned they did not know the personal details of their risk contacts. During this period, individuals attending the same gathering were classified as HRC in case the attack rate was high.

## Key performance indicators (KPIs)

One of the KPIs for contact tracing according to ECDC and WHO is the proportion of new cases arising from known contacts [13, 14]. These cases have previously received information regarding their potential exposure to the virus, as well as information on testing and quarantine, and we defined them as 'known' index cases. In contrast, we defined 'new' index cases as individuals that tested positive for SARS-CoV-2 but were not previously identified as a risk contact. We investigated differences between these 'new' and 'known' index cases in terms of the number of traced contacts and secondary cases, in order to describe the impact of contact tracing on onward transmission. Due to the lack of reliable household information for the entire study period, we did not initially make a distinction between household and non-household contacts. Since the end of May 2021, the identifier for traced HRC included an indicator of household status. Therefore, for the period from June 2021 to December 2021, we can divide

HRC into household and non-household contacts. Delay distributions were constructed to investigate how quickly contact tracing was performed. The time between the index case receiving their positive test result and getting contacted by the tracing system gives an indication of how quickly the contact tracing system operated. To assess the intervention as a whole, we also calculated the delay between symptom onset of an index case and its SARS-CoV-2 test, as well as between symptom onset and the receipt of the result of that test, which gives an indication of how quickly individuals are tested after showing symptoms and how quickly the test results are available.

## Contact and transmission patterns

Based on the available data sources, we constructed a contact line list in which each traced contact with known NRN was linked to their index case. A transmission line list was conceived by linking infected contacts to their reported index case if positive tests occurred in a 21-day time span, assuming that transmission could have occurred in either direction [16]. If infection of an HRC or LRC was confirmed more than 21 days before or after their index case, transmission between these individuals was deemed very unlikely and the pair was only kept in the contact line list. We had no information on whether non-positive contacts tested negative or were not tested at all. Sequencing information was available to identify variants of concern (VoC) for a limited number of cases. When sequencing information was available for both individuals in a potential transmission pair, but different VoCs were identified, this pair was removed from the transmission line list.

To investigate contact patterns and transmission dynamics over time, age-specific matrices were constructed. We assumed that transmission would have occurred from index to contact. Contact matrices represent the number of contacts between an index in age group $i$ and a contact in age group $j$, divided by the total number of index cases in age group $i$. The transmission matrices represent the number of infections in age group $j$ that can be linked to an index case in age group $i$, divided by the total number of infections in age group $j$. Since only limited sequencing information was available, we analyzed contact and transmission matrices for specific months during which each VoC was the dominant circulating strain [17]. In December 2020, there was no circulation of a VoC yet. In April and July 2021, there was circulation of the Alpha and Delta VoC, respectively. The Omicron VoC started circulating in Belgium in December 2021, though it did not yet dominate the Delta VoC [17].

Another important transmission characteristic is the serial interval, defined as the time between symptom onset in the index case and symptom onset in their secondary case, which gives an indication of the speed of transmission. We calculated the serial interval over time as the monthly moving average by date of the positive test of the index case (i.e. forward serial interval), assuming that the index case was the source of infection and randomly selecting one index case when there were multiple possibilities. Individuals for whom the timing of symptom onset was not available, because they did not report it or were asymptomatic, were excluded from this calculation. The serial interval was restricted to lie within the biologically plausible interval of -5 to 21 days [18].

## Ethical approval

This study has been approved by the Flemish Government's Department of Care (GE0–1GDF2IA-WT/1GD305/20069780). It was carried out in accordance with international ethical standards (Declaration of Helsinki 1964). It was conducted in accordance with the General Data Protection Regulation (GDPR) and a data processing agreement was concluded between the Flemish Government's Department of Care and the Universities of Antwerp and Hasselt.

All personal data were encrypted by the Flemish Government's Department of Care before being made available for analysis.

## Results

Data extracted on 25 May 2022 included 1,004,694 index cases (987,210 unique individuals) identified between September 2020 and December 2021 (S1 Fig, S1 Table). We were able to link 1,092,985 traced contacts (917,102 unique individuals) to 416,645 unique index cases. In the remainder of this work, we will focus on HRC, since these make up the majority of traced contacts (94.9%). Among the index cases for which at least one HRC was registered in the tracing system, the average number of registered HRC was 2.8 (IQR 1—4). Overall, 76.4% of the registered HRC were effectively traced (S2 Fig). Among index cases linked to at least one traced HRC, the average number of traced HRC was 2.5 (IQR 1—3). During the period from June 2021 to December 2021, 46.6% of all traced HRC were defined as household contacts. For about 30% of the traced HRC, the NRN was missing, resulting in a substantial part of traced HRC that could not be included in these analyses. Furthermore, 1.5% of the traced HRC were linked more than once to the same index case, 8% were linked to an index case without a known identifier, and 3% were linked to an index case not included in the available data.

### 'Known' index cases

From September 2020 to December 2021, 19.1% of all index cases had been previously identified as a risk contact (i.e. 'known' index cases), of which 96.7% as HRC. Starting at less than 10% during September–November 2020, the proportion of 'known' index cases fluctuated around 20% for the remaining study period, being highest (24.8%) during May–June 2021 and lowest (18.9%) during July–August 2021 (Fig 1a, Table 1). The proportion of 'known' index cases was highest among 0–17 year-olds and lowest among those above 65 years old. Changes in the proportion of 'known' index cases over time followed a similar trend within each age group (S3 Fig).

### High-risk contacts

Index cases that were not previously identified as a risk contact (i.e. 'new' index cases) were associated with a significantly higher average number of traced HRC compared to 'known' index cases (Fig 1b, Table 1). A drop in the number of traced HRC is observed during November–December 2021, with a smaller but still significant difference between 'known' and 'new' index cases during that period. In general, the average number of traced HRC was higher for 'new' index cases regardless of household status, although the difference with 'known' index cases was less pronounced for household HRC (S4 Fig).

### Secondary cases and SAR

The average number of secondary cases was relatively stable for 'known' index cases until September 2021, and significantly lower compared to the number of secondary cases linked to 'new' index cases (Fig 1c, Table 1). In addition to the absolute number of secondary cases, we also compared the secondary attack rate (SAR), representing the number of secondary cases among all contacts. For contacts that could be linked to multiple index cases (13.7% of all HRC), we randomly selected one index case when calculating the SAR. The SAR was significantly higher for 'new' index cases from December 2020 until August 2021 (Fig 1d, Table 1). Although the number of traced HRC in the household remained relatively stable over time, an

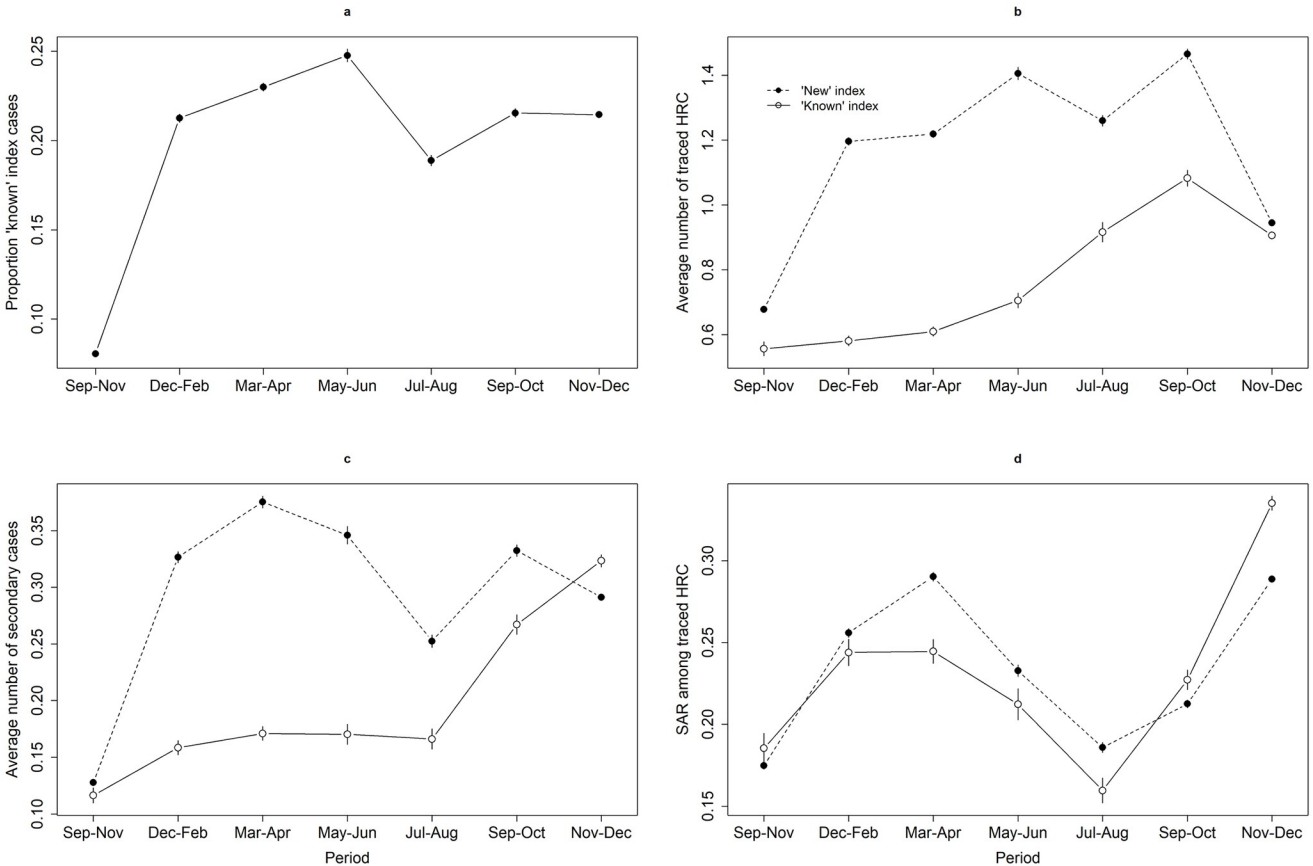

**Fig 1. Evolution of key performance indicators.** Evolution in the (a) proportion of index cases that were previously identified as a risk contact, (b) average number of traced high-risk contacts (HRC) for 'new' and 'known' index cases, (c) average number of secondary cases for 'new' and 'known' index cases, and (d) secondary attack rate (SAR) among traced HRC of 'new' and 'known' index cases. Vertical bars represent the 95% confidence interval.

increase was observed in the number of secondary cases and SAR for household contacts (S4 Fig).

Of the 189,900 index cases contacted between September and November 2020, 8.1% had previously been identified as a risk contact. This proportion increased between 1 September and 12 October, after which it decreased until its lowest point around 31 October 2020 before increasing again. During the period around 31 October 2020, there was no substantial difference in the number of traced HRC and secondary cases between 'known' and 'new' index cases, while it was observed that 'known' index cases were associated with fewer HRC and secondary cases for the remaining period (S5 Fig).

## Delay distributions

The average time between receiving the positive test result and being contacted by the tracing system was 1.3 days, with 90% of index cases being contacted within 3 days of their test result. The time between positive test and tracing increased between November 2020 and March 2021 compared to the remaining study period (Fig 2). For symptomatic index cases (data available from November 2020 onward), the average time between symptom onset and taking a SARS-CoV-2 test was 2.3 days, with 90% of the index cases tested within 5 days after symptom

**Table 1. Key performance indicators.** Mean and 95% confidence interval over time. Bold notation indicates a significant difference between 'known' and 'new' index cases.

| | Sep-Nov 2020 | Dec 2020—Feb 2021 | Mar-Apr 2021 | May-Jun 2021 | Jul-Aug 2021 | Sep-Oct 2021 | Nov-Dec 2021 |
|---|---|---|---|---|---|---|---|
| **Proportion of 'known' index cases** | 0.081 (0.079—0.082) | 0.213 (0.210—0.215) | 0.230 (0.228—0.232) | 0.248 (0.244—0.251) | 0.189 (0.186—0.192) | 0.215 (0.213—0.218) | 0.215 (0.213—0.216) |
| **Average number of traced HRC** | | | | | | | |
| 'Known' index | **0.556** (0.535—0.578) | **0.581** (0.566—0.596) | **0.610** (0.596—0.624) | **0.706** (0.682—0.729) | **0.916** (0.886—0.947) | **1.083** (1.058—1.107) | **0.906** (0.896—0.917) |
| 'New' index | **0.678** (0.671—0.686) | **1.196** (1.185—1.208) | **1.219** (1.208—1.230) | **1.405** (1.386—1.425) | **1.260** (1.243—1.277) | **1.465** (1.450—1.481) | **0.945** (0.939—0.951) |
| **Average number of secondary cases** | | | | | | | |
| 'Known' index | **0.116** (0.110—0.123) | **0.159** (0.152—0.165) | **0.171** (0.165—0.177) | **0.170** (0.161—0.179) | **0.166** (0.157—0.175) | **0.267** (0.258—0.276) | **0.323** (0.318—0.329) |
| 'New' index | **0.128** (0.126—0.130) | **0.327** (0.322—0.332) | **0.375** (0.370—0.380) | **0.346** (0.338—0.354) | **0.253** (0.247—0.258) | **0.332** (0.327—0.338) | **0.291** (0.289—0.294) |
| **SAR among traced HRC** | | | | | | | |
| 'Known' index | 0.185 (0.176—0.195) | **0.244** (0.236—0.252) | **0.245** (0.237—0.252) | **0.212** (0.203—0.222) | **0.160** (0.152—0.167) | **0.227** (0.221—0.233) | **0.335** (0.331—0.340) |
| 'New' index | 0.175 (0.173—0.177) | **0.256** (0.253—0.259) | **0.290** (0.288—0.293) | **0.233** (0.229—0.236) | **0.186** (0.183—0.189) | **0.213** (0.210—0.215) | **0.289** (0.287—0.291) |

onset. The average time between symptom onset and their positive test result was 2.9 days, with 90% of the index cases receiving their test results within 6 days after symptom onset. These delays were fairly stable throughout the investigated period (Fig 2).

## Contact and transmission patterns

Fig 3 shows the age-specific contact and transmission matrices for December 2020, April 2021, July 2021, and December 2021. During each period, we observed an assortative relation in both the contact and transmission matrices, i.e. most contacts and transmission occurred within age groups. In July 2021, the average number of contacts between individuals aged 20 to 29 years was highest, which is also reflected in the transmission matrices. An increase in transmission from 10–19 year-olds to 40–49 year-olds is observed during July 2021 (when the Delta VoC was dominant in Belgium), while the average number of traced contacts between these age groups remained relatively stable over time. Similarly, in December 2021 (i.e. rise of the Omicron VoC), we observe an increase in transmission from 0–9 year-olds to 30–39 year-olds. The serial interval was shortened over time, in line with previous studies [19, 20]. In addition, we found that the serial interval was shorter for 'known' index cases compared to 'new' index cases from July 2021 onward (Fig 4).

## Discussion

In this study, we found that SARS-CoV-2 confirmed cases that had previously been identified as a risk contact generally reported fewer HRC and were associated with fewer secondary infections, as well as a lower secondary attack rate among their HRC. These results reflect the mandatory quarantine for identified risk contacts, but the lower attack rates also suggest that people who were informed of their exposure to an infected person were more careful with the contacts they did have. For both 'known' and 'new' index cases, we observed an increasing

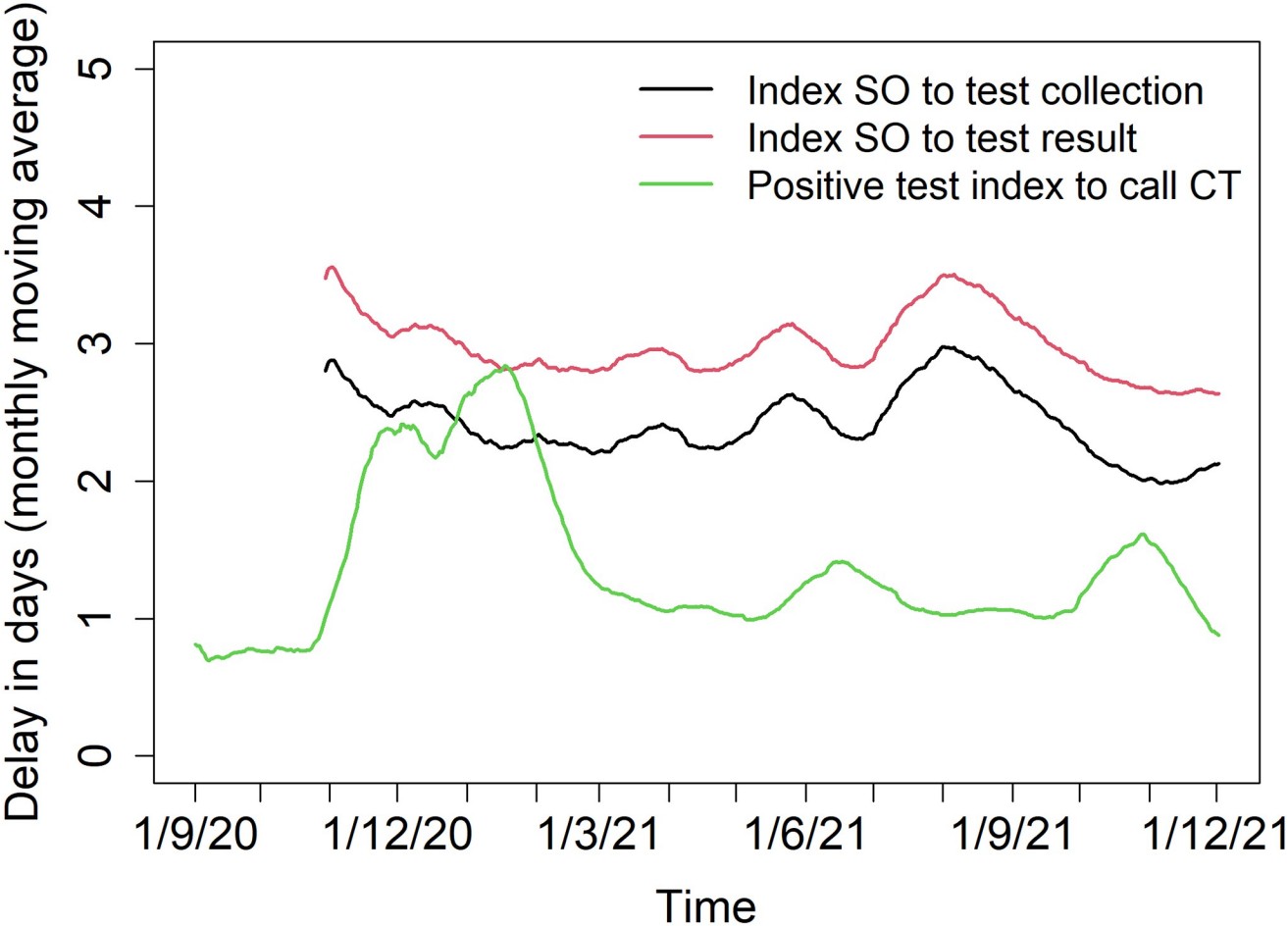

**Fig 2. Delay distributions.** Delay distributions for the time between index symptom onset and test collection (black), between index symptom onset and test result (red), and between positive test index and the call from the tracing center (green).

trend in the number of traced HRC, which may be explained by the relaxation of NPIs over time. Delay distributions indicate that the contact tracing system operated fairly quickly during the majority of the study period.

Results from the period September–November 2020 indicate that KPIs obtained during periods of high burden on the healthcare system should be interpreted with caution due to possible changes in testing and quarantine strategies that disturb the standard data flow on which these indicators are based. For example, during the period September–November 2020, we observed a decrease in the proportion of index cases that were previously identified as a risk contact, coinciding with a strong peak in SARS-CoV-2 incidence in the population resulting in changes in testing and tracing strategies. The change in contact tracing strategy after the summer of 2021 may also have impacted the KPIs obtained since September 2021, since index cases that had attended a gathering were possibly linked to many more registered and tested HRC compared to other index cases.

A study evaluating the contact tracing system in Catalonia, Spain, found an increase in the proportion of index cases that had already been identified as a contact from 34% in May 2020 to 58% in November 2020 [21]. This is considerably higher than what we found for Belgium, with the proportion being at most 25% during the period May–June 2021. Furthermore, in

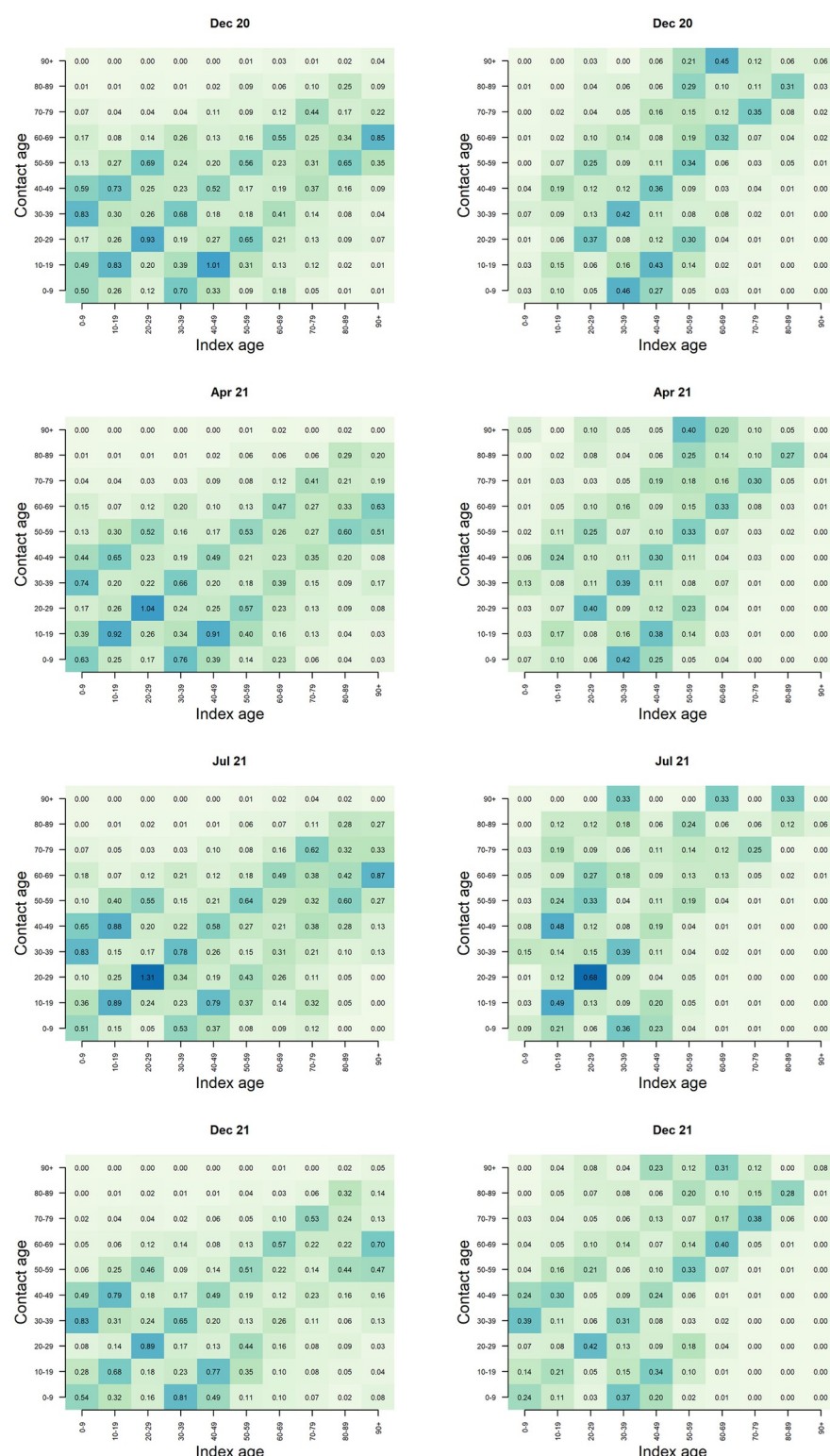

**Fig 3. Contact and transmission patterns.** Age-specific contact (left) and transmission (right) matrix for December 2020, April 2021 (i.e. Alpha VOC), July 2021 (i.e. Delta VOC), and December 2021 (i.e. Omicron VOC).

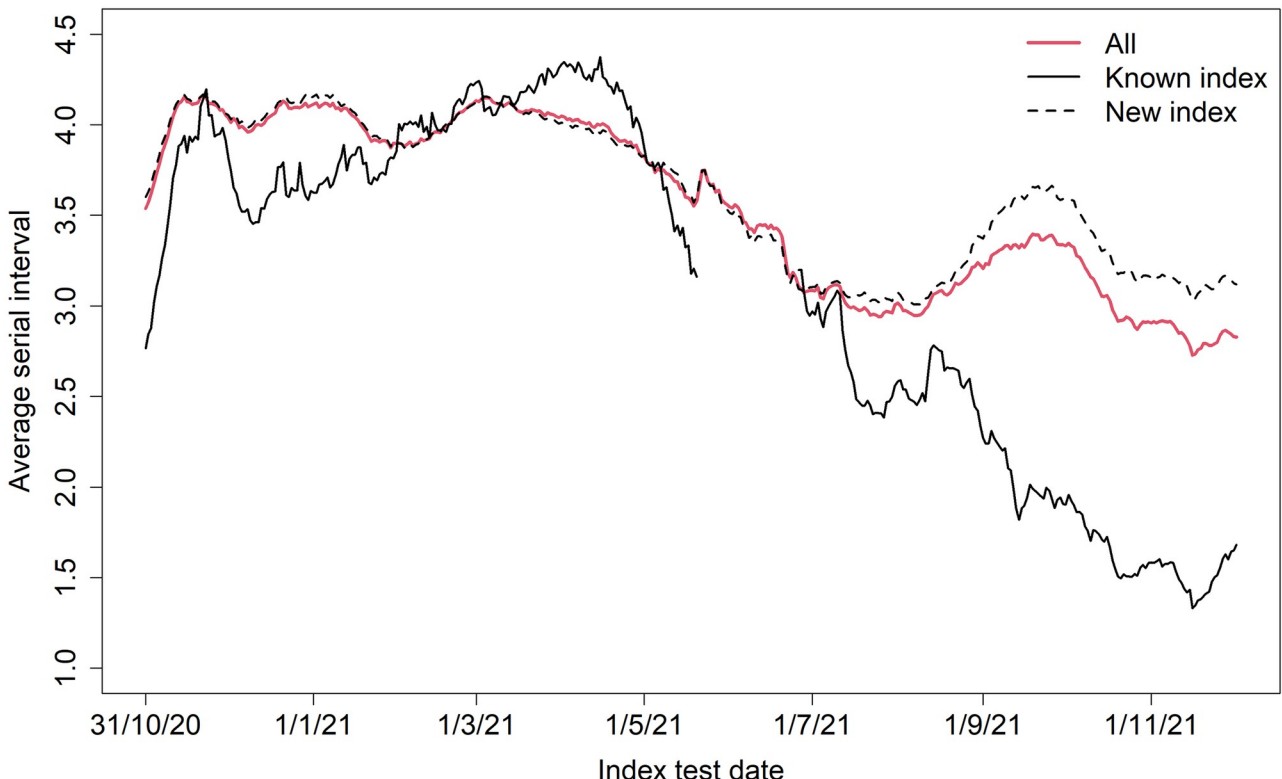

**Fig 4. Serial interval.** Monthly moving average empirical serial interval (in days; 116,453 transmission pairs included; 15,190 transmission pairs with 'known' index and 101,263 with 'new' index included).

our study only 42.2% of the index cases were linked to at least one traced HRC, while in Catalonia 67.1% of index cases reported at least one close contact, of which 99.8% were effectively traced. A study in Portugal conducted between 1 March and 30 April 2020 compared the distribution of secondary cases between index cases that were previously identified as a close contact or returned from affected areas and those that had not been subjected to contact tracing or quarantine measures prior to diagnosis [22]. In line with our results, they found that 'known' index cases were associated with a lower number of close contacts. On the contrary, they did not find a difference in the number of secondary cases or SAR between 'known' and 'new' index cases. Country-specific differences in contact tracing setup, operator training, inquiries made, and the absolute number of confirmed cases to process are likely to have an impact on the effectiveness of contact tracing.

In addition to breaking transmission chains, contact tracing provides an important source of information regarding transmission characteristics. The observed increase in transmission from 10–19 year-olds to 40–49 year-olds during circulation of the Delta VoC, and similarly from 0–9 year-olds to 30–39 year-olds during the rise of the Omicron VoC, indicates an increased transmission potential of children when these VoCs were circulating. Another contributing factor could be the increased vaccination coverage in adults in 2021, resulting in reduced susceptibility, which also implies a reduced potential to transmit the disease given the reduced chance of infection [23]. At the end of July 2021, 60% of the general population of Belgium was fully vaccinated, while at the end of December 2021, 75% of the general population was fully vaccinated and 38% of the adult population had received a booster vaccine [24]. A

previous study that linked Belgian contact tracing data with vaccine uptake reported that only 36% of the index cases were fully vaccinated by the end of September 2021 [25].

The shorter serial interval observed for 'known' index cases could be explained by the mandatory quarantine endorsed by the tracing, which limits their social contacts and as a result lowers their probability of transmitting the virus later in their infectious period [19]. Furthermore, previous studies have reported a shorter serial interval for Omicron compared to Delta VoC, suggesting faster transmission [26, 27]. In that sense, a shorter serial interval jeopardizes the impact of contact tracing since the turnaround time needs to be reduced, which may not be achievable. This may explain the increased SAR observed in November–December 2021 when the Omicron VoC started circulating.

In addition to operational challenges and disease characteristics, such as asymptomatic infections, the effectiveness of contact tracing also depends on testing policy and compliance, the cooperation of index cases, and the imposed and adopted quarantine and isolation measures. Some index cases might, willingly or unwillingly, not report all of their contacts. This could explain why the number of traced HRC observed in our study is lower than the average of 4 contacts that has been found based on social contact surveys [28]. Even when including only those index cases that were linked to at least one traced HRC, the average was only 2.5 (IQR 1—3) HRC. A possible explanation is that duplicate contacts were usually linked to only one index case to avoid contacting the same individuals more than once. Additionally, individual contacts that were traced as part of a collectivity are not included in the available data. Although the most important thing is to prevent onward transmission, for research and surveillance purposes, it is beneficial to record all individual contacts for each index case. In this way, more comprehensive contact and transmission networks can be reconstructed, which is important for estimating epidemiological characteristics such as the reproduction number and generation interval [28–30].

This study is limited by the lack of information on negative clinical tests. As such, we did not know whether risk contacts that did not reappear as index cases were not infected or not tested. Additionally, the availability of the NRN was crucial to assess conversion from contact to index case, resulting in a large part of traced HRC that could not be included in these analyses due to missing NRN. This may have led to an underestimation of the KPIs presented here. In addition, whereas all 'known' cases have been previously traced and quarantined, 'new' cases are likely to include some cases who were quarantined but not registered as HRC, especially if they were household contacts of an index case. From the available data, the directionality of transmission cannot be assessed. It is possible that an index case was infected by one of their traced contacts, but was the first individual to show symptoms and/or to be contacted by the contact tracing. Due to specific guidelines for testing and contact tracing in collectivities such as nursing homes and schools, the available data are also not fully representative of young children and the elderly population because the majority of their contacts have been traced and followed up by local medical services. Therefore, transmission within these age groups may be underrepresented in our transmission matrices. This may also explain the low proportion of index cases aged over 85 years, as well as the low proportion of 'known' index cases among those aged over 65 years. An additional explanation may be the earlier vaccination of elderly, protecting them against infection.

In order to estimate an absolute number of prevented infections, a reference period or region would be needed in which similar NPIs were in place, the same variants were circulating, and, most of all, with contact data being collected without informing risk contacts on their potential exposure, explaining measures to be taken, and motivating contacts to comply. We did not have such reference data, and hence we are restricted to analyzing and reporting relative differences between index cases that were previously informed of their exposure by the

contact tracing, and index cases that were not previously contacted by the contact tracing operators. Alternatively, an individual-based model can be used to quantify the effectiveness in terms of what would have happened if no contact tracing was in place. Such a simulation study has been performed for Belgium early in the pandemic, and future research could extend this model to include contact tracing scenarios as they were implemented in real-life [5].

## Conclusion

While we did not quantify the effectiveness of contact tracing in all its dimensions, we show that contact tracing in Belgium from September 2020 to December 2021 has been effective in reducing onward transmission. Slowing down the epidemic by contact investigation is of major importance in order to prevent the collapse of healthcare systems [31]. This study also shows that in times of high burden on the healthcare system, contact tracing KPIs should be interpreted with caution in light of changing testing and quarantine strategies. The KPIs used in this study, i.e. the proportion of 'known' index cases, number of contacts, secondary cases, SAR, and delay distributions, can be used to continuously monitor the performance and impact of contact tracing. In addition, the data obtained by contact tracing allow to investigate transmission patterns by individual characteristics of the index case, such that contact tracing can be prioritized to individuals with a high contribution to transmission at a certain point in time.

## Supporting information

**S1 Fig. Number of index cases.** Number of index cases that were contacted during the period from September 2020 to December 2021 by age group, with an overview of the most influential control measures regarding social contacts.
(TIF)

**S2 Fig. Traced high-risk contacts.** The proportion of registered high-risk contacts (HRC) that was effectively traced, over time (monthly moving average).
(TIF)

**S3 Fig. 'Known' index by age group.** Evolution in the proportion of 'known' index cases by age group. Vertical bars represent the 95% confidence interval.
(TIF)

**S4 Fig. Key performance indicators by household status.** Evolution by household status in the (a) average number of traced HRC for 'new' and 'known' index cases, (b) average number of secondary cases for 'new' and 'known' index cases, and (c) secondary attack rate (SAR) among traced HRC of 'new' and 'known' index cases.
(TIF)

**S5 Fig. Key performance indicators fall 2020.** Evolution in the (a) proportion of index cases that were previously identified as a risk contact, (b) average number of traced high-risk con-tacts (HRC) for 'new' and 'known' index cases, (c) average number of secondary cases for 'new' and 'known' index cases, and (d) secondary attack rate (SAR) among traced HRC of 'new' and 'known' index cases, for the period from September to November 2020.
(TIF)

**S1 Table. Index case characteristics.** Proportion of all index cases by province and age group.
(PDF)

## Acknowledgments

The authors are grateful for the many interesting discussions within the Controletoren Consortium including members of Hasselt University, Ghent University, KU Leuven, and University of Antwerp.

## Author Contributions

**Conceptualization:** Cécile Kremer, Lander Willem, Christel Faes, Niel Hens.

**Data curation:** Cécile Kremer, Lander Willem.

**Formal analysis:** Cécile Kremer.

**Investigation:** Cécile Kremer, Lander Willem, Jorden Boone, Naïma Hammami, Niel Hens.

**Methodology:** Cécile Kremer, Lander Willem, Niel Hens.

**Project administration:** Christel Faes, Niel Hens.

**Resources:** Cécile Kremer, Lander Willem, Jorden Boone, Naïma Hammami.

**Software:** Cécile Kremer.

**Supervision:** Christel Faes, Niel Hens.

**Validation:** Jorden Boone, Wouter Arrazola de Oñate, Naïma Hammami.

**Visualization:** Cécile Kremer.

**Writing – original draft:** Cécile Kremer, Lander Willem, Niel Hens.

**Writing – review & editing:** Cécile Kremer, Lander Willem, Jorden Boone, Wouter Arrazola de Oñate, Naïma Hammami, Christel Faes, Niel Hens.

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
