## [Decision Letter · Decision Letter 0]

11 Jul 2023

PONE-D-22-27415Key performance indicators of COVID-19 contact tracing in Belgium from September 2020 to December 2021PLOS ONE

Dear Dr. Kremer,

Thank you for submitting your manuscript to PLOS ONE. I apologize for the delay in the editorial process. Your paper was evaluated by an expert in the field and myself. Though the topic may be important, there are many points that need to be clarified. Please read the comments carefully and address the issues accordingly.

We look forward to receiving your revised manuscript.

Kind regards,

Tomohiko Ai, M.D., Ph.D.

Academic Editor

PLOS ONE

Journal Requirements:

“Naima Hammami is an employee of the the Flemish Agency for Care and Health and Jorden Boone has been working as a consultant for the agency during the COVID-19 pandemic.

Niel Hens declares that the Universities of Antwerp and Hasselt have received funding for advisory boards and research projects of MSD, GSK, JnJ, Pfizer outside the proposed work. Niel Hens has not received any personal remuneration related to this work.

The other authors declare that they have no competing interests.”

We note that one or more of the authors are employed by a commercial company: Flemish Agency for Care and Health

Please include both an updated Funding Statement and Competing Interests Statement in your cover letter. We will change the online submission form on your behalf

Reviewers' comments:

Reviewer's Responses to Questions

**Comments to the Author**

1. Is the manuscript technically sound, and do the data support the conclusions?

Reviewer #1: Partly

2. Has the statistical analysis been performed appropriately and rigorously? 

Reviewer #1: No

3. Have the authors made all data underlying the findings in their manuscript fully available?

Reviewer #1: Yes

4. Is the manuscript presented in an intelligible fashion and written in standard English?

Reviewer #1: No

5. Review Comments to the Author

Reviewer #1: Overall, the authors address a relevant topic in the field of epidemiological surveillance and intervention in public health. The key performance indicators are well suited, and the division between cases that were previously identified as close contacts and those that were not yet known is adequate. Although other papers have already suggested the effectiveness of contact tracing, there is an abundance of modelling studies and a lack of community and real-world studies. The former are dependent on a plethora of premises that often fail to represent reality, whereas the latter reflect reality with all its complexities, from which I believe this manuscript could be relevant.

The manuscript, however, is poorly designed. The introduction is far too large, and could be reduced to half the size without losing any of its relevant points. In fact, its length – a problem applicable to the entire paper – draws the reader off the idea the authors are trying to communicate.

The title and the abstract focus mainly on the effectiveness of contact tracing. That should be the main aim of the paper, as given by the objectives in the end of the introduction. However, when one reaches the Results and Discussion section, it is mingled with secondary analysis (age and close contacts) which are more dubious, and definitely less relevant. The authors focus too much on the statistical presentation of secondary analysis, without much consideration for the message they are trying to pass and the epidemiological implications of their choices. Also, the authors rely on their figures heavily and provide few data on the results text section, which is a major limitation. The inclusion of 6 figures, with no tables (such as baseline characteristics) suggests that authors were unable to focus their message on the essential. In fact, after reading the manuscript several times, it seems that authors changed their focus midway, which denotes a frail conception.

Notes by parts

Abstract

“(…) fewer close contacts as well as fewer secondary cases and a lower secondary attack rate (…)”: the absence of quantitative data is a major limitation. Even in the abstract, major results ought to be given.

The conclusion that “…individuals aware of their exposure to SARS-CoV-2 seemed more reserved in their social contact behavior” is, at best, an euphemism for the fact that they were placed under quarantine and had less contacts to report, which is the point of contact tracing and quarantine. This is no direct consequence of the results here provided and should not be included.

Introduction:

Lines 8-9: “When restrictions were relieved, countries implemented contact tracing…”. As I understand from the manuscript, this was how Belgium tackled the pandemic; however, in many settings that was not so. In my country, contact tracing was operational prior to the pandemic (for other infectious diseases) and was implemented prior to, during, and after all lockdowns.

Lines 12-14: conceptually, other problems like asymptomatic cases transmitting disease, short incubation periods, acute disease, are all characteristics affecting the effectiveness. Also, a reference is missing.

Lines 20-26: Superfluous. I understand the relevance of the cited papers, yet the previous sentences already established that potential benefits of contact tracing. The authors may opt to include reference 9 in the end of line 20.

Lines 27-34: Same thing. Interesting results to support the effectiveness of contact tracing, but which add nothing to what was previously established. Also, in the first line “It is not trivial…” it seems to this reviewer that there would be no need to compare regions with and without contact tracing, because you are comparing cases subjected to contact tracing to “new” cases. That distinction is already made. Therefore, this section is redundant.

Lines 42-onwards: These are methods. It includes setting, time window and context. I would move this to the methods sections (albeit in fewer sentences) and conclude only with the objectives

Lines 55-56: “We investigated several key indicators and discuss their opportunities and pitfalls”. Meaningless sentence. It is relevant to know which key indicators were selected, which the authors provide in the next sentence.

Lines 59-60: attack rate?

Methods:

Lines 75-80: There is no specification as to what type of measures were implemented for high and low-risk contacts. One assumes that high-risk patients were placed under quarantined, whereas low-risk were either subjected to active surveillance, or were disregarded due to lack of resources. The end sentence (between brackets) is redundant and should be deleted.

Lines 81-87: Although the authors take a lot of consideration to distinguish between reported, registered and traced contacts. I understand that registered contacts were not necessarily traced, which happened frequently for many reasons. However, it is not clear to this reviewer the difference between reported and registered. Was it possible for a case to report 10 contacts, but the interviewer would only register 8 in the system?

Lines 99-100. Other (referenced) papers opted to exclude these cases and contacts.

Line 126 and line 146: there appears to be a contradiction. The transmission line is said to be constructed for cases and their reported index case if the test was positive 21 days before or after the index case. In the end of the paragraph, it is claimed that the serial interval was constrained to -5 to 21 days. Is this referring to different steps of the process?

Results:

The presentation of results is absurd, and needs a complete overhaul.

Lines 143-136: Most likely estimating the speed of contact tracing (which is relevant) rather than transmission.

Line 168: To use percentages to provide numbers larger than 100% is a poor choice to present data. Additionally, I can’t figure where this number comes from. 416 645 unique cases represent 41.5% of all included cases. Even if I may understand there is a dot/comma missing in virtually every data provided, still the number is not correct.

Lines 168-170: If the result you have is summarized in a figure, with no more than a sentence claiming the same as the legend, then it is superfluous. It is an interesting image to present in a workshop or seminar, eventually, but it adds nothing here.

Line 178: It seems like a proportion, and hence could not be 191%. Assuming 19.1%, this is different from the result given in the abstract (20%).

Line 180-181: The proportion seems stable, except for the first point. Rephrase.

Lines: 188-189: Discussion!

Figure 2: Where are the confidence intervals? Are these results significant?

Line 199: 137%. Same issue.

Line 200: 4 points below vs 3 points. Virtually similar. Dubious interpretation.

Line 207: S4 figure is absolutely redundant, even as a supplement.

Line 208-220: This paragraph is a mix of methods, results and discussion. Needs complete rewriting.

Figure 3: Redundant. Doesn’t seem to add to the discussion.

Line 234-242: No clear reason for the stratification per month. Given the information which is provided in the discussion, would it be not more useful to stratify per variant? Also, the differential proportion of vaccination during the time period is never taken into account, statistically. Another option was to stratify per strategy, as given in the first paragraph of the discussion.

Lines 243-250: It is unclear to this reviewer the relevance of this analysis. Given the bias acknowledged by failing to include older people, the results are inconsequential. I would delete this sub-analysis.

Line 336: One third of HRC were excluded. This is tremendous, and should have 1) led the authors to focus more on the effectiveness of contact tracing and quarantine, which is relegated to a secondary position after the introduction, and 2) to provide some comparison between included and excluded HRC, to analyze whether the included are representative of the population and, if not, how not so.

Discussion

Lines 266-267: Individuals attending the same gathering were classified as HRC <- highly relevant data that should be given in the methods section!

Lines 293-295: See previous comment.

Line 301-306: This suggests that analysis should have been stratified by variant, which is meaningful, rather than monthly, which is comparing completely different scenarios.

Lines 307-317: It is not clear why this paragraph is included, at all.

Limitations should note that there may be a differential misclassification of ‘exposure’, as always happens in cases like this: all ‘known’ cases were definitively previously traced, whereas ‘new’ cases are likely to include a few cases who were quarantined, but whose quarantine was unknown to the researchers/public health authorities.

6. PLOS authors have the option to publish the peer review history of their article (what does this mean?). If published, this will include your full peer review and any attached files.

Reviewer #1: No

---

## [Author Response · Author response to Decision Letter 0]

5 Sep 2023

Please find the Response to Reviewers in the uploaded Word document.

---

## [Editor Report · Decision Letter 1]

20 Sep 2023

Key performance indicators of COVID-19 contact tracing in Belgium from September 2020 to December 2021

PONE-D-22-27415R1

Dear Dr. Kremer,

We’re pleased to inform you that your manuscript has been judged scientifically suitable for publication and will be formally accepted for publication once it meets all outstanding technical requirements.

Kind regards,

Tomohiko Ai, M.D., Ph.D.

Academic Editor

PLOS ONE
---

## [Editor Report · Acceptance letter]

12 Oct 2023

PONE-D-22-27415R1 

Key performance indicators of COVID-19 contact tracing in Belgium from September 2020 to December 2021 

Dear Dr. Kremer:

I'm pleased to inform you that your manuscript has been deemed suitable for publication in PLOS ONE. Congratulations! Your manuscript is now with our production department. 

Kind regards, 

on behalf of

Dr. Tomohiko Ai 

Academic Editor

PLOS ONE